**Data Availability Statement:** Data cannot be shared publicly due to regulatory constraints. Data

# Effect of an intensive cervical traction protocol on mid-term disability and pain in patients with cervical radiculopathy: An exploratory, prospective, observational pilot study

**Thomas Rulleau**[1,2,3]☯*, **Sophie Abeille**[4]☯, **Lydie Pastor**[2,3], **Lucie Planche**[1], **Pascale Allary**[2], **Catherine Chapeleau**[2], **Chloé Moreau**[1], **Grégoire Cormier**[3], **Michel Caulier**[3]

1 Unité de Recherche Clinique, CHD Vendée, La Roche-sur-Yon, France, 2 Kinésithérapie, CHD Vendée, La Roche-sur-Yon, France, 3 Service de Rhumatologie, CHD Vendée, La Roche-sur-Yon, France, 4 Centre de Médecine Physique et de Réadaptation (CMPR) Côte d'Amour, Saint Nazaire, France

☯ These authors contributed equally to this work.
* thomas.rulleau@chd-vendee.fr

## Abstract

### Background

Cervical radiculopathy is a relatively common and disabling condition involving local pain in the neck region and pain that radiates into the upper limb. Recent data suggest that cervical traction may effectively reduce disability and pain, with a dose-response relationship. The main aim of this study was therefore to evaluate the mid-term effect of an intensive cervical traction protocol for patients with cervical radiculopathy on disability, and to compare the effects with those reported by non-intensive protocols in the literature.

### Methods

We conducted a prospective open observational study of 36 patients referred by their general practitioner for symptoms suggestive of cervical radiculopathy. All patients under-went the same treatment: a 30-minute cervical traction protocol, twice a day, for five consecutive days. The main objective was the evaluation of disability at 3 months. We evaluated at baseline (D1), the end of the protocol (D5) and at mid-term (M3) disability, cervical pain, radiating pain, pain on motor imagery, presence of neuropathic pain and medication consumption. The primary outcome was the proportion of patients for whom the Neck Disability Index improved by more than the minimum clinically important difference of 7 points by M3.

### Results

Thirty-six patients were included in this study. The Neck Disability Index improved by more than the minimum clinically important difference in 48.3% at M3. Mean Neck Disability Index (p < .001), mean cervical VAS (p < .001), mean radiating VAS (p < .001), and mean VAS for imagined lateral flexion and rotation (p < .002) improved significantly from D1 to D5 and

are available from the Institutional Data Access Committee of the CHD Vendée Clinical Research Unit for researchers who meet strict confidential data access criteria. Data cannot be shared publicly because of the regulatory constraints and patient's consent form. Data are available from the CHDVendée's scientific committee, and its president: Dr Couvrat, at gregoire.couvrat@chd-vendee.fr for researchers who meet the criteria for access to confidential data.

**Funding:** The author(s) received no specific funding for this work.

**Competing interests:** The authors have declared that no competing interests exist.

from D1 to M3. Consumption of medication reduced at each time point. The proportion of patients with neuropathic pain reduced from 61.1% at D1 to 33.3% at D5 and 48.3% at M3.

## Conclusion

Disability reduced by more than the minimum clinically important difference in almost half of the participants following the intensive traction protocol. These results are encouraging and suggest that this complex condition can be treated with relatively simple methods.

## Introduction

Cervical radiculopathy is a relatively common and disabling condition [1] involving local pain in the neck region and pain that radiates into the upper limb. It is usually caused by a disc herniation or another space-occupying lesion that causes impingement and/or inflammation of the cervical nerve root [1, 2]. The prevalence of cervical radiculopathy is reported to be 3.5 per 1000 people [3], and the annual incidence varies from 83 cases to 210 cases per 100,000 people, with a peak from 50 to 59 years [4, 5]. The diagnosis may be confirmed by magnetic resonance imaging, electrophysiological testing (e.g. nerve conduction velocity tests or electromyography) or clinical examination (neck pain with referred pain to the arm, upper extremity paresthesia or numbness and signs of nerve root compression) [3].

A review of the natural history, clinical course, and prognostic factors of symptomatic cervical disc herniations with radiculopathy found that substantial improvements tend to occur within the first 4 to 6 months after onset [6]. Time to complete recovery ranges from 2 to 3 years [6], therefore this condition is associated with high costs due to repeated or prolonged sick leave, multiple evaluations (such as imaging) and multiple treatments, (including physiotherapy, surgery and medication) [7].

Treatment approaches are varied but the main aim of current treatments is to reduce pain and disability in the short-term, as well as to prevent recurrence in the long-term [6, 8]. Low quality evidence suggests that surgery may provide faster pain relief than physiotherapy or hard collar immobilization in patients with cervical radiculopathy or myelopathy [9]; but there is little or no difference in the long-term [9]. Conservative treatments involve strengthening, stretching, manual therapy, massage, medication and traction [10]. Cervical traction induces a separation of the vertebral bodies, movement of the facet joints, expansion of the intervertebral foramen, and stretching of soft tissues [3, 11]. A recent study found no difference between manual cervical traction and manual therapy and a combination of these techniques on pain, disability and cervical mobility [12], however a meta-analysis found a dose-response relationship of traction on these outcomes [3]. Romeo et al (2018) conducted a review and meta-analysis of five studies that compared the effects of traction with another treatment on pain and disability in adults with cervical radiculopathy [3] (Table 3). The meta-analysis found that studies that included more sessions and longer traction times resulted in better pain and disability outcomes.

Thus, according to the literature, cervical traction appears to reduce pain and disability in cervical radiculopathy [3]. Furthermore, this treatment is low cost. In order to further investigate the apparent dose-response relationship found by Romeo et al., we wished evaluate the impact of an intensive traction program provided over a short period of time in patients with cervical radiculopathy. Our working hypothesis was that a more intensive protocol (ten 30-minute traction sessions provided over 5 days with massage) could shorten the recovery time and reduce the risk of chronicity, accelerate return to work and reduce the overall costs

associated with cervical radiculopathy. The main aim of this study was therefore to evaluate the mid-term effect of an intensive cervical traction protocol for patients with cervical radiculopathy on disability, and to compare these effects with data from non-intensive protocols in the literature.

## Method

### Details of study design

We conducted a single center (Departmental Hospital Center—La Roche sur Yon site (Vendée, France), prospective, observational study that tested an intensive cervical traction protocol (see below for details). All patients received the same treatment.

### Population

Patients were referred to our rheumatology department for assessment and treatment of cervical radiculopathy by their general practitioner. On admission, a clinical examination was performed by a rheumatologist who diagnosed cervical radiculopathy if the patient had pain radiating to the arm with motor and/or sensory dysfunction. These criteria were also used as study selection criteria by Romeo et al. for their meta-analysis. Symptoms reduced in six patients before beginning the traction protocol and the rheumatologist did not diagnose cervical radiculopathy (i.e. neck pain no longer radiated into the arm and there was no motor or sensory dysfunction), therefore they were not enrolled in the rehabilitation program. All other patients were enrolled in a rehabilitation program that included cervical traction.

**Inclusion and non inclusion criteria.** Patients over 18 years of age, who were enrolled in the cervical traction program (as part of usual care in our center), who could be followed at 3 months, and who had given their non-opposition, were included.

Patients who were under guardianship, unable to understand the protocol, diagnosed with myelopathy, cancer, arterial pathologies, fracture-dislocation or spinal infection were not included. Cervical spine x-rays are systematically performed in our center to eliminate contra-indications to manual treatment.

### Intervention/issue of interest (exposure)

To propose an intensive cervical traction, thirty-minute traction sessions were performed twice daily for 5 days (total of 10 sessions) by a physiotherapist. Participants were positioned in supine [13] on a flat bed. A manual traction test was performed to ensure that traction did not provoke any pain or unwanted sensations. Participants were asked to keep their gaze forwards throughout the traction to avoid muscle contraction by oculocephalogyric coupling [14]. The mechanical traction was applied at 45˚ from the horizontal plane formed by the bed [15] (Fig 1).

Mechanical traction was set to 5% of the participant's body weight on day 1 (unless not tolerated) and increased to reach 10% on the 5th day, as pain allowed, and without exceeding 12kg [14]. Within a session, the load was applied progressively over 5 minutes and pain was monitored using a VAS [16]. The target weight was then maintained for 20 minutes and then reduced gradually over the last 5 minutes. The patient then lay still for 10 minutes before getting up [14, 17]. A physiotherapist was present during the entire first session and the patients were provided with a call bell for the other sessions.

On the mornings of days 2, 3 and 4 (before the traction), the patients also underwent 15 minutes of massage. A physiotherapist performed effleurage, kneading, muscle tension release and stretching techniques as required, with the patient sitting in a massage chair [18].

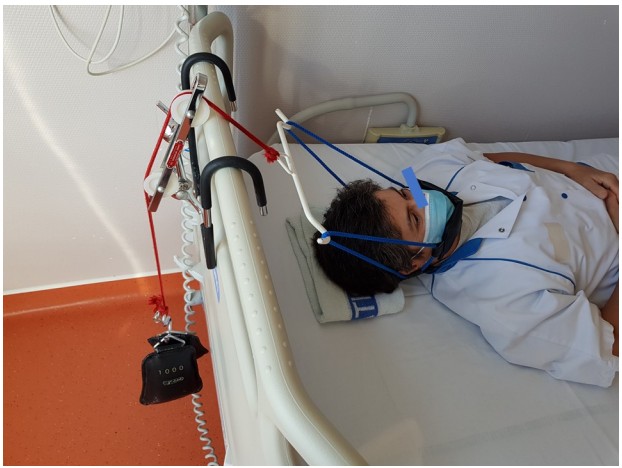

**Fig 1. Cervical traction set-up.**

## Evaluation

Physiotherapists and rheumatologists who were trained in the outcomes used in the study conducted the baseline assessments on the first morning, before the traction began (D1) and on Day 5 (D5). Another pre-trained physiotherapist or a resident rehabilitation physician performed the assessments at 3 months +/- 15 days (M3).

**Primary outcome.** The primary outcome was the percentage of patients with a clinically important reduction in disability by M3. The Neck Disability Index combines pain intensity, personal care, lifting, reading, headaches, concentration, work, driving, sleeping and recreation [19, 20]. It has excellent reliability, internal consistency, and validity [20, 21]. It is composed of 10 questions and can be self-completed by the patient. The maximal score is 50 points, and higher scores show higher levels of disability. The minimum clinically important difference for patients with cervical radiculopathy is 7 points [19]. We therefore considered a decrease of 7 or more points in the Neck Disability Index between Day 1 (before first traction) and Mid-term (Month 3, M3)as a clinically important improvement in disability.

**Secondary outcomes.** *Disability*. Mean Neck Disability Index score was evaluated at D1, D5 and M3.

*Pain*. Local (cervical) and radiating (into the upper limb) pain were evaluated using a visual analogue scale (VAS) at D1, D5 and M3 [16, 22].

*Central sensitization*. Pain during motor imagery of flexion, rotation and lateral flexion was evaluated using a VAS [23, 24] at D1, D5 and M3.

*Neuropathic pain*. Evaluated using the Neuropathic Pain Diagnostic Questionnaire (DN4) at D1, D4 and M3. A score ≥4 indicates neuropathic pain [25]. The proportion of patients with neuropathic pain was compared at each time point (D1, D5 and M3).

*Consumption of medication*. The type and amount of medication taken for the cervical radiculopathy was recorded at each time point.

*Sick leave*. The number of days of sick leave since the onset of symptoms at hospitalization (D1) at M3 was reported.

*Nerve recovery*. Two signs of nerve recovery were evaluated at D1, D5 and M3: the deep tendon (muscle stretch) reflexes [26] and the length and mobility of various components of the nervous system (Upper Limb Nerve Tension Test 1a) [3, 27].

## Monitoring of adverse events

Adverse events were monitored by a physician, a nurse and a physiotherapist. In particular we monitored pain and signs of nerve compression, hypotension when sitting up, dizziness and tinnitus, headaches, nausea, fainting, muscle damage, progression of local cervical pain to radiating pain or to loss of nerve conduction.

## Ethics

The study 2017-A02004-49 began on April 9, 2018, after being granted ethical approval (Comité de Protection des Personnes Ile de France XI, approval number 18018), and was conducted in accordance with the Helsinki convention. In accordance with the legislation, oral consent was obtained status at the time of their inclusion in the study.

## Statistical analysis

**Sample size determination.** We did not perform a sample size calculation since our aim was to conduct a pilot observational study. The number of patients was based on the predicted inclusion capacity over one year.

**Primary outcome.** The percentage of patients with an improvement in disability at 3 months (decrease of at least 7 points on the Neck Disability Index between D1 and M3) was calculated.

**Secondary outcomes.** Change in outcomes over time was evaluated using linear mixed models that took into account a random subject effect. Changes in Neck Disability Index score and pain were described by means and standard deviations at each time point. The number and proportion of patients with a positive ULNT1a was calculated at each time point. The analysis of consumption of medication, number of days of sick leave and presence of deep tendon reflexes at each time point was descriptive (numbers and percentages).

The statistical analysis was performed with "R" and the significance level was set at $p \leq .05$.

# Results

## Population

No patients refused to participate, thus thirty-six were included. Mean age was $51.1 \pm 12.1$ years (the distribution is illustrated in Fig 2), 20 (55.5%) were female and mean BMI was 25.4 ($\pm 3.38$) kg/m$^2$; 33.3% were unemployed or retired, 23.8% were sedentary workers, and 42.9% had jobs that involved physical work. Mean symptom onset was $22.6 \pm 31.1$ months before the start of treatment. All patients had cervical radiculopathy as diagnosed by the rheumatologist. Only 29 patients could be evaluated at M3: 3 were unavailable as they were undergoing surgery and 4 were lost to follow-up.

## Primary outcome: Neck Disability Index

Neck Disability Index score reduced by more than the minimum clinically important difference of 7 points in 48.3% of patients (Fig 3).

## Other outcomes

**Disability.** Neck Disability Index score decreased significantly over time ($p < .001$) (Table 1), from D1 to D5 ($p < .001$) and D1 to M3 ($p < .001$); no difference was found between D5 and M3 ($p = .44$). No difference in improvement was found according to the duration of symptoms ($p > .239$).

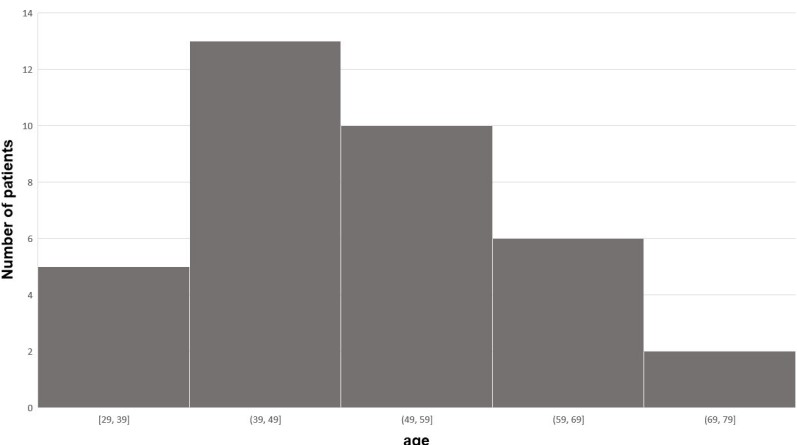

**Fig 2. Frequency of patients by age.**

**Pain.** The cervical VAS score decreased significantly over time (p < .001) (Table 1), from D1 to D5 (p < .001) and D1 to M3 (p < .01); no difference was found between D5 and M3 (p = .52).

Radiating VAS score decreased significantly over time (p < .001) (Table 1), from D1 to D5 (p < .001), and D1 to M3 (p < .001); no difference was found between D5 and M3 (p = .36).

The VAS score of the imagined lateral flexion and rotation movements decreased over time (p < .01), from D1 to D5 (p < .01) (Table 1). The imagined lateral flexion VAS score also decreased from D1 to M3 (p < .001) and the imagined rotation VAS score decreased from D1 to M3 (p = .043). There was no change in VAS score for imagined flexion over time (D1-D5 (p = .22) and D1-M3 (p>.21)). For all imagined movements, there were no differences for any movement between D5-M3 (p>.91).

61.1% of patients had neuropathic pain at D1, 33.3% at D5 and 48.3% at M3 (Fig 4).

**Consumption of medication (Fig 5).** Use of non-steroidal anti-inflammatory drugs (NSAIDs) reduced from 22.2% at D1 to 11.4% at D5 and then 0% at M3. Use of antidepressants was 11.1 at D1, 14.3 at D5 and 10.3% at M3. Use of antiepileptics reduced from 13.9 at D1 to 14.3% at D5 and 10.3% at M3. Use of class 1 analgesics increased from 66.7 then 74.3% at D5 and then reduced to 37.9 at M3. Use of class 2 analgesics reduced from 19.4 to 17.1 then

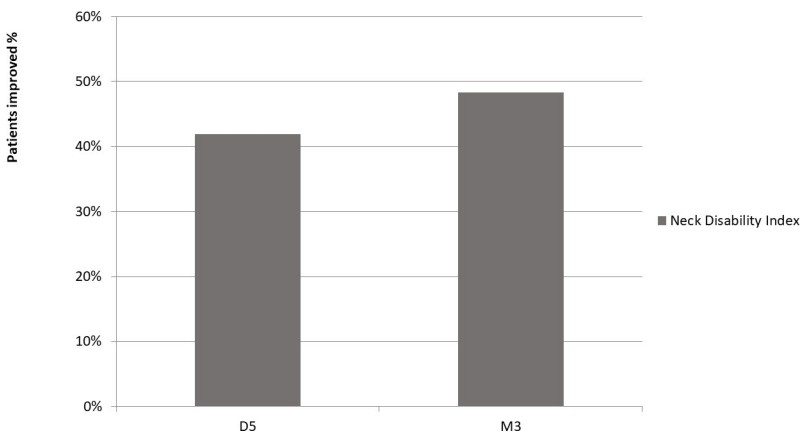

**Fig 3. Proportion of patients with improvement in disability above the MCID.**

**Table 1. Results and comparison of disability and pain evolution between D1, D5 and M3.**

|  | D1 | D5 | M3 | p (D1-D5) | p (D5-M3) | p (D1-M3) |
|---|---|---|---|---|---|---|
| Neck Disability Index/50; (sd)) | 19.1 (6.3) | 14.8 (8.1) | 12.1 (7.9) | < .001 | p>.438 | < .001 |
| Cervical pain VAS/100; (sd) | 36.7 (23.3) | 15.1 (24.4) | 18.7 (19.5) | < .001 | p = .523 | < .01 |
| Arm pain VAS/100; (sd) | 41.5 (25.0) | 16.0 (17.8) | 22.0 (22.7) | < .001 | p = .364 | < .001 |
| Cervical flexion VAS/100; (sd) | 20.0 (23.6) | 12.7 (18.5) | 14.8 (23.9) | = .213 | = 0.475 | = .915 |
| Imagined lateral flexion VAS/100; (sd) | 28.8 (23.6) | 13.7 (18.1) | 14.2 (21.6) | < .001 | = .93 | < .01 |
| Imagined rotation VAS/100; (sd) | 27.5 (23.9) | 12.7 (20.1) | 14.0 (22.0) | < .01 | = .97 | = .043 |

10.3%. Use of class 3 analgesics reduced from 13.9% at D1 to 8.6% at D5 then 6.9% at M3. Use of corticosteroids reduced from 19.4 to 2.9 then 0%.

**Sick leave.** The average duration of sick leave was 125.0 (±331.6) days before inclusion in the protocol.

**Deep tendon reflexes.** Reflexes were absent in 31.7% at D1, 26.3% at D5 and 10.7% M3.

**ULNT1a.** The proportion of participants with a positive ULNT1a reduced from 60.6% at D1 to 36.4% at D5 (p = .066) and 36.0% at M3 (p = .11) with no difference between D5 and M3 (p = .99).

# Discussion

## Main findings of the present study

This prospective observational study showed a clinically important reduction in disability (> 7 points on the Neck Disability Index, [19]) in 48.3% of the patients with cervical radiculopathy 3 months after beginning an intensive cervical traction protocol. Furthermore, this improvement was not related to the duration of symptoms. Disability, local and radiating pain, central sensitization and neuropathic pain, medication consumption, and neurological status (Reflexes and ULNT1a) were all significantly improved at the end of the protocol (D5) and remained so at the 3-month follow up. Although this study was un-controlled, we believe that in view 1) of the chronicity of the symptoms (mean duration 20.4 months, SD = 31.2), and 2) the short duration of the protocol (effects found after only 5 days), these results can be attributed to the effects of the traction protocol.

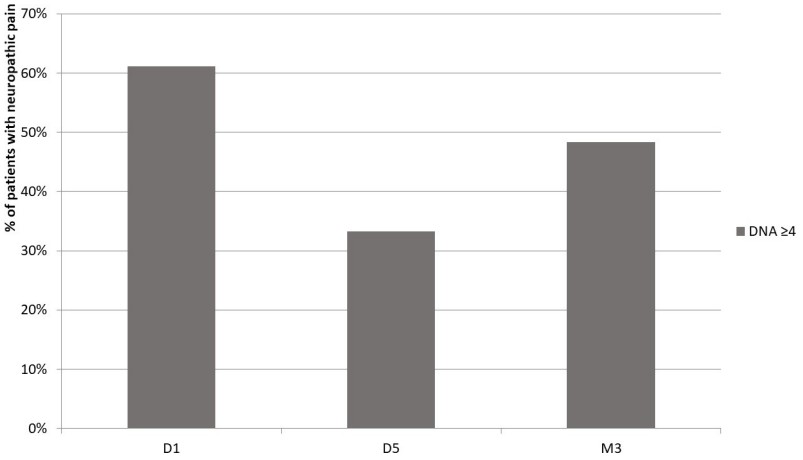

**Fig 4. Proportion of patients with neuropathic pain.**

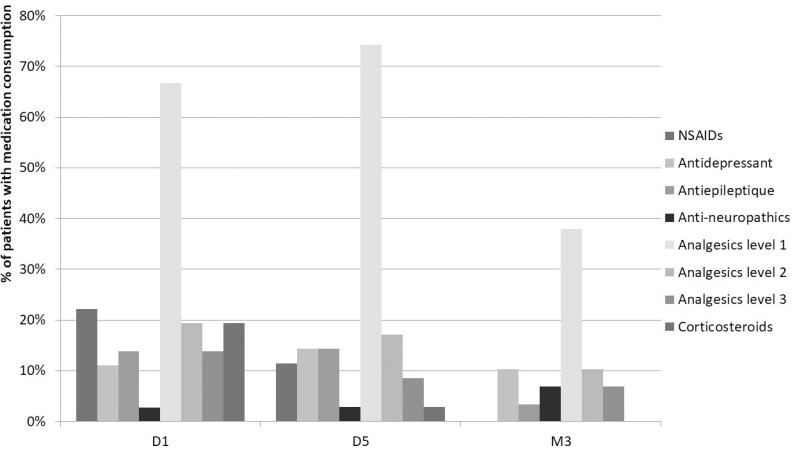

**Fig 5. Change in medication consumption.**

The primary outcome of the present study (proportion of patients with a change in Neck Disability Index greater than the minimum clinically important difference) could not be directly compared with other studies since they did not use this outcome. The secondary disability outcome (change in Neck Disability Index at 3 months) is relatively common in the literature, however comparison with similar studies [12, 28–31] was limited by the use of different methods: the Neck Disability Index can be rated out of 50 or 100 ([29], please note that Table 2 shows all results converted to a scale out of 50 points), however the authors did not always specify their choice (based on the values reported, we made the assumption that [30] used a scale out of 100 points), and one study used a VAS to evaluate disability [29]. Four

**Table 2. Comparison of effects of traction in the literature.**

| | Disability | | | Cervical pain | | | Arm pain | | | Pain (no distinction between arm and cervical) | | |
|---|---|---|---|---|---|---|---|---|---|---|---|---|
| | baseline | T1 | T2 | baseline | T1 | T2 | baseline | T1 | T2 | baseline | T1 | T2 |
| Jellad et al. (2009) B group | 48,1 (VAS) | 23,2 (VAS) | No Data | 58,3 (VAS) | 33,3 (VAS) | No Data | 66 (VAS) | 31,9 (VAS) | No Data | No Data | No Data | No Data |
| Moustafa et Diab (2014) A group | 18,8 (NDI) | 13,5 (NDI) | 17,3 (NDI) | 6,5 NPRS | 4,6 NPRS | 6,30 NPRS | 6,1 NPRS | 4,2 NPRS | 5,8 NPRS | No Data | No Data | No Data |
| Fritz et al. (2014) Mech tract group | 15,4 (NDI) | 4,8 (NDI) | 8,7 (NDI) | 3,9 NPRS | 1,00 NPRS | 1,1 NPRS | 4,3 NPRS | 1,4 NPRS | 0,9 NPRS | No Data | No Data | No Data |
| Aydin et Yazicioglu (2012) Traction group | No Data | No Data | No Data | No Data | No Data | No Data | No Data | No Data | No Data | 69,2 (VAS) | 24,6 (VAS) | No Data |
| Young et al. (2009) MTEXTraction Group | 19,8 (NDI) | 14 (NDI) | 11,1 (NDI) | No Data | No Data | No Data | No Data | No Data | No Data | 6,3 NPRS | 4,20 NPRS | 3,4 NPRS |
| Afzal et Al. (2019) Traction group | 22,4 (NDI) | 10,6 (NDI) | No Data | No Data | No Data | No Data | No Data | No Data | No Data | 7,5 NPRS | 3,08 NPRS | No Data |
| Results for patients | 19,1 (NDI) | 14,8 (NDI) | 12,1 (NDI) | 36,7 (VAS) | 15,1 (VAS) | 18,7 (VAS) | 41,5 (VAS) | 16,0 (VAS) | 22,0 (VAS) | No Data | No Data | No Data |

T1 = just at the end of the protocol. T2 = mid-term evaluation; VAS: Visual Analogue Scale; NPRS: Numeric Pain Rate Scale; NDI: Neck Disability Index, The NDI has been scaled to 50 to allow comparison.

of the 5 studies shown in Table 2 used the Neck Disability Index at mid-term and thus we were able to compare our results with these. Three of these studies (Moustafa et al., Fritz et al. and Young et al.) found reductions in disability of a similar order to the present study (around 7 points) at mid-term. Importantly, the improvement in disability was greater than that which would be expected for the normal course of the disease [6], suggesting it was indeed due to the treatment.

The reductions in cervical and radiating arm pain were also above the minimum clinically important differences for these variables: 8.1/100 for cervical pain and 10.4/100 for radiating arm pain [32]. The baseline level of pain was lower in our study compared with the studies shown in Table 2. This is likely due to differences in care provided in different countries. Furthermore, comparison is hindered by different methods of pain measurement: some used the Numeric Pain Rating Scale (NPRS) [12, 28, 30, 31] and not all distinguished cervical and radiating pain [12, 31, 33]. Among the studies that used a VAS to measure pain, the reduction varied from 25 to 45 points. In the present study the reduction was around 20 points. This can be considered similar in view of the fact the baseline levels were lower, thus there was less potential for reduction.

The evaluation of the effect of traction on central sensitization was novel in this study. It has previously been shown that in the case of chronic pain, the flow and integration of neural activity within the pain matrix [34, 35] is altered [24]. These changes can be indirectly evaluated by assessing pain produced during motor imagery [24], which is normally painless. The baseline measures of pain during cervical flexion, extension and rotation demonstrated the presence of central sensitization in the patients included, likely due to their long history of symptoms [24]. This pain reduced by approximately 50% following the traction and, importantly, the reduction was maintained 3 months later. The positive evolution of pain during imaginary movement, despite the long duration of the symptoms, may indicate that central remodeling occurred [24]. This fact is important because patients with central sensitization seems to have more severe pain, poorer general health-related quality of life, and greater levels of pain-related disability, depression, and anxiety [36]. There is some evidence that low-back pain treatment reverses abnormal brain function [37, 38] but, to our knowledge, this is the first time that this reversal has been demonstrated following traction for cervical radiculopathy.

To our knowledge, this is the only study to have evaluated the effect of cervical traction on neuropathic pain in the case of cervical radiculopathy. The percentage of patients with a score above 4 points on the DN4 reduced from 61.1% to 48.3% at M3, indicating that 8 patients no longer had neuropathic pain [25, 39]. This is very interesting because neuropathic pain is highly challenging to treat. Most currently available treatments are only moderately effective and have side effects that limit their use (e.g. medications) [40]. The results of the present study demonstrate that, in some patients, neuropathic pain may be reduced by an intense, short and specific protocol.

Few studies have evaluated drug consumption as a treatment outcome for cervical radiculopathy [3]. However, this parameter is important for two reasons. Firstly, an increase in drug intake could favorably influence the primary outcome (and vice versa). Secondly, a reduction in drug consumption is an important indicator of treatment success, as well as being important for the patients' overall health. Furthermore, by M3, none of the patients were taking NSAIDs or corticosteroids. The use of level 2 and 3 analgesics, as well as anti-epileptic drugs to reduce neuropathic pain, was also decreased at D5 and M3. Only the prescription of antidepressants did not change. This was not unexpected since all patients had chronic pain, however it is unlikely to have affected the improvement in disability and pain.

We evaluated variables relating to nerve recovery because of the specific decompression effect of cervical traction on nerve tissue [3, 11]. The ULNT1a evaluates the nerve's ability to slide and elongate [41] and thus provides information regarding the biomechanics of the peripheral nerve tissue [41]. Reflexes, on the other hand, provide an indication of nerve conduction [42] which appeared to increase slightly by the end of the traction protocol and more substantially by M3. However, there was only a trend towards an improvement in the biomechanics of the nerve. The traction thus appeared to have a greater effect on nerve conduction through decompression than on any inflammation present in the nerve, as has previously been reported for other manual therapy techniques [43].

## Implications and explanation of findings

The improvement in all the clinical variables (Neck Disability Index, intensity and type of pain, drug consumption, as well as ULNT1a and deep tendon reflexes) following the traction protocol is positive. In their review, Romeo et al. (2018) suggested that a higher number of traction sessions led to a greater improvement in outcomes. Our protocol involved a number of sessions (10 sessions) that was comparable with studies in the literature (7–15 sessions), however we provided these sessions over a shorter time-frame (5 days). This more intensive protocol has several advantages. Firstly, the recovery time may have been accelerated (5 days). This may reduce the duration of sick leave which would in turn reduce the costs associated with cervical radiculopathy.

## Study limitations

The main limitation of this work was the number of participants lost to follow-up, however this is quite typical in this population [28] (Table 3). It is due to the fact some patients had returned to work and did not attend their final consultation, while conservative treatment had failed for several others and surgery was planned before the end of the follow-up.

## Future directions

Randomized, controlled trials comparing traction with sham traction are now required to fully determine the effectiveness of cervical traction on cervical radiculopathy. The optimal traction modalities, such as pull angle and weight, also need to be determined.

**Table 3. Comparison of protocol conditions across studies.**

| Study | N = | Lost to follow up | Number of sessions | Duration of the protocol (in weeks) | Mean no. sessions per week | Duration of the traction session (in minutes) | Intensity of the traction (in kilograms) |
|---|---|---|---|---|---|---|---|
| Jellad et al. (2009) | 39 | 0 | 12 | 4 | 3 | 25+25 | 5 to 12kg |
| Moustafa et Diab (2014) | 216 | 27 | 12 | 4 | 3 | 20 | 9,1 to 15,9kg |
| Fritz et al. (2014) | 86 | 6mo = 22<br>12mo = 32 | 10 | 4 | 2,5 | 15 | 3,6 to 9,1kg |
| Aydin et Yazicioglu (2012) | 27 | No data | 15 | 3 | 5 | 20 | 5 to 20kg |
| Young et al. (2009) | 81 | 8 | 7 | 4 | 1,75 | 15 | 9,1 to 15,9kg |
| Afzal et Al. (2019) | 40 | 1 | 9 | 3 | 3 | 10 | 10 to 15% of body weight |
| Results | 36 | 7 (including 3 for surgery) | 10 | 1 | 10 | 30 | 5 to 10% of body weight, under 12kg |

## Conclusion

Disability reduced by more than the minimum clinically important difference in almost half the participants following the intensive traction protocol. In addition, cervical and radiating arm pain, pain with imagined movements and neuropathic pain also improved. Furthermore, all the secondary outcomes also improved. These results are encouraging and suggest that this complex condition can be treated with relatively simple methods.

## Acknowledgments

We would like to thank Johanna Robertson for translation and constructive criticism.

## Author Contributions

**Conceptualization:** Thomas Rulleau, Sophie Abeille, Lucie Planche, Grégoire Cormier, Michel Caulier.

**Formal analysis:** Thomas Rulleau.

**Funding acquisition:** Sophie Abeille.

**Investigation:** Thomas Rulleau, Sophie Abeille, Lydie Pastor, Pascale Allary, Catherine Chapeleau, Chloé Moreau, Grégoire Cormier, Michel Caulier.

**Methodology:** Thomas Rulleau, Sophie Abeille, Lucie Planche, Grégoire Cormier, Michel Caulier.

**Supervision:** Thomas Rulleau, Grégoire Cormier, Michel Caulier.

**Validation:** Sophie Abeille.

**Writing – original draft:** Thomas Rulleau, Sophie Abeille, Lucie Planche, Grégoire Cormier, Michel Caulier.

**Writing – review & editing:** Thomas Rulleau, Sophie Abeille, Lucie Planche, Grégoire Cormier, Michel Caulier.

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
