## [Decision Letter · Decision Letter 0]

4 May 2021

PONE-D-21-09986

Effect of an intensive cervical traction protocol on mid-term disability and pain in patients with cervical radiculopathy: an exploratory, prospective, observational pilot study

PLOS ONE

Dear Dr. Rulleau,

Thank you for submitting your manuscript to PLOS ONE. After careful consideration, we feel that it has merit but does not fully meet PLOS ONE’s publication criteria as it currently stands. Therefore, we invite you to submit a revised version of the manuscript that addresses the points raised during the review process.

We look forward to receiving your revised manuscript.

Kind regards,

Panagiotis Kerezoudis, M.D., M.S.

Academic Editor

PLOS ONE

Journal Requirements:

For additional information about PLOS ONE ethical requirements for human subjects research, please refer to " ext-link-type="uri" xlink:type="simple">http://journals.plos.org/plosone/s/submission-guidelines#loc-human-subjects-research."

6. Please include your tables as part of your main manuscript and remove the individual files. Please note that supplementary tables (should remain/ be uploaded) as separate "supporting information" files.

Reviewers' comments:

Reviewer's Responses to Questions

**Comments to the Author**

1. Is the manuscript technically sound, and do the data support the conclusions?

Reviewer #1: No

Reviewer #2: Yes

Reviewer #3: Yes

Reviewer #4: Yes

2. Has the statistical analysis been performed appropriately and rigorously? 

Reviewer #1: Yes

Reviewer #2: Yes

Reviewer #3: Yes

Reviewer #4: Yes

3. Have the authors made all data underlying the findings in their manuscript fully available?

Reviewer #1: Yes

Reviewer #2: Yes

Reviewer #3: Yes

Reviewer #4: Yes

4. Is the manuscript presented in an intelligible fashion and written in standard English?

Reviewer #1: No

Reviewer #2: Yes

Reviewer #3: Yes

Reviewer #4: Yes

5. Review Comments to the Author

Reviewer #1: Roulleau T and colleagues here present the results of their study named “Effect of an intensive cervical traction protocol on mid-term disability and pain in patients with cervical radiculopathy: an exploratory, prospective, observational pilot study”. In this work the authors conducted an observational study of 36 patients referred by their general practitioner for symptoms suggestive of cervical radiculopathy, submitting them to a specific cervical traction protocol, analyzing post-protocol symptoms and signs also through the help of Neck Disability Index. They found that disability was significantly reduced in almost half of the participants, suggesting such protocol as an alternative to longer traction protocols, at the comparison of whom the present protocol appeared similar in terms of efficacy.

Although the work is sufficiently well written and clear from a stylistic point of view, I honestly feel it suffers from heavy scientific rigor issues that definitely affect its potential scientific interest.

I list here my major concerns:

- Introduction section is really too long, please re-write it in a more summarized manner;

- Introduction section, lines 63-66 definitely need references, if available. Moreover, the entire paragraph should be re-written specifying for which pathologies such considerations are referred. Are the authors speaking about cervical radiculopathy due to disk herniation or stenosis or tumors etc.? Anyway, I would be very interested in reading some reference that states that intervention is not indicated for cervical radiculopathy in cervical disk herniations with pain resistant to medications.

- Material and methods. Was cervical radiculopathy diagnosed just with clinical examination, without performing MR or CT imaging to patients? No specific radiological control before submitting them to cervical traction? Really? It appears, at least, risky.

- No neurosurgeons nor ortophaedic surgeons appeared to be involved in the study. Hence, how can patients be informed about their condition and supposed diagnosis? From the reading of this work it seems that, simply, patients with cervical radiculopathy that come to your clinic are directly submitted to cervical traction, without any diagnosis nor alternative treatment proposal; If it’s not, re-write the section entirely, please;

- English written language may beneficiate from a mother-tongue speaker review.

Reviewer #2: To Whom It May Concern,

This is an exceptional paper. However, there are a few data points that would add to the paper. Those points are a graphic (histogram) representation of the age breakdown in intervals. We know that older patients are more prone to degenerative spondylosis however, it would be helpful to know where the majority of the study population resides in age distribution. Also, in the methods section I could not discern the exact method used to apply traction. For example, did you all use Gardner Wells Tongs or strap system? Otherwise, well written paper.

Reviewer #3: First, I would like to express sincere gratitude to get an opportunity to review the manuscript. The endeavor of the authors is appreciated. The authors have studied effect of intensive cervical traction with respect to mid-term disability and pain in patients with cervical radiculopathy. However, there is some scope for its improvement.

Specific comments:

1. There are some contradictory statements in the paper. For example, as per methods section 36 patients were enrolled in the study whereas as per results section of main text 42 patients were enrolled. Kindly explain.

2. The study design is not clear. Was there any control arm? The conclusion states there was some comparison done. The details of study design need to be well described.

3. Kindly define mid-term effect and intensive cervical traction.

4. It would be better if study design were clearly mentioned in title, abstract and main text.

5. Methods sections is supposed to be core of any study. Here, methods section contains inadequate information. For example, following components for methods section need to be well described.

i. Details of Study design

ii. Setting

iii. Sampling technique

iv. Participant

v. Primary and secondary outcome variables with working definition

vi. Intervention/issue of interest (exposure)

vii. Comparison (if done)

viii. Ethics and end point

ix. Statistical analysis

6. Kindly re-frame the references as per the guidelines to authors in the home page of journal. Further, most of the studies included in the reference section seem to be the studies published more than 10 years back. If possible, please add recent studies.

7. The tables need to be re-organized. For example, table 2 appears first. Further, kindly provide study size in captions. It would be better to frame all the tables in homogenous format.

Section wise comments

1. Abstract contains inadequate information.

a. Mention the design of the study clearly.

b. Methods section is not well described.

c. Kindly start results section with baseline information of participants.

2. The introduction is lengthy. Please DELETE INFORMATION UNRELATED TO OBJECTIVE so that the section is short and sweet. Kindly shorten this section and delete unrelated information. Kindly focus on three elements of introduction

a. What is known about the topic? (Background)

b. What is not known? (The research problem)

c. Why the study was done? (Justification)

3. Methods section determines the results. Kindly follow the checklist. Further, kindly focus on three basic elements of methods section.

a. How the study was designed?

b. How the study was carried out?

c. How the data were analyzed?

4. The discussion section needs to be described scientifically. Kindly frame it along the following lines:

a. Main findings of the present study

b. Comparison with other studies

c. Implication and explanation of findings

d. Strengths and limitations

e. Conclusion, recommendation and future directions

5. Conclusion needs to provide answers for each objective clearly in a sentence or two.

Reviewer #4: Journal of PLOS ONE

27 April 2021

Manuscript No: PONE-D-21-09986_reviewer

“Effect of an intensive cervical traction protocol on mid-term disability and pain in patients with cervical radiculopathy: an exploratory, prospective, observational pilot study”

Dear editorial Teams

In my opinion the manuscript has well written. However, I have several suggestions that I think would improve the manuscript:

a) The Introduction and the Discussion sections are long and redundant.

b) Additional descriptions as NDI (line 155- 162) should be summarized, and ets.

c) Researchers stated that “ forty-two patients were included” and “Only 29 patients could be evaluated at M3: 3 were unavailable as they were undergoing surgery and 4 were lost to follow-up” 42= 29+ 3+ 4+ X? please explain the other 6 patients.

d) Please, explain the type of traction used.

Parisa Azimi, MD,

6. PLOS authors have the option to publish the peer review history of their article (what does this mean?). If published, this will include your full peer review and any attached files.

Reviewer #1: No

Reviewer #2: No

Reviewer #3: **Yes: **Dr. Satish Prasad Barnawal

Reviewer #4: No

---

## [Author Response · Author response to Decision Letter 0]

10 Jun 2021

We would like to thank the editor and reviewers for taking the time to evaluate our work. We hope that the changes made will meet their expectations. 

Reviewer #1:

Roulleau T and colleagues here present the results of their study named “Effect of an intensive cervical traction protocol on mid-term disability and pain in patients with cervical radiculopathy: an exploratory, prospective, observational pilot study”. In this work the authors conducted an observational study of 36 patients referred by their general practitioner for symptoms suggestive of cervical radiculopathy, submitting them to a specific cervical traction protocol, analyzing post-protocol symptoms and signs also through the help of Neck Disability Index. They found that disability was significantly reduced in almost half of the participants, suggesting such protocol as an alternative to longer traction protocols, at the comparison of whom the present protocol appeared similar in terms of efficacy.

Although the work is sufficiently well written and clear from a stylistic point of view, I honestly feel it suffers from heavy scientific rigor issues that definitely affect its potential scientific interest.

I list here my major concerns:

- Introduction section is really too long, please re-write it in a more summarized manner;

We thank reviewer 1 for this suggestion and have modified as suggested

- Introduction section, lines 63-66 definitely need references, if available. Moreover, the entire paragraph should be re-written specifying for which pathologies such considerations are referred. Are the authors speaking about cervical radiculopathy due to disk herniation or stenosis or tumors etc.? Anyway, I would be very interested in reading some reference that states that intervention is not indicated for cervical radiculopathy in cervical disk herniations with pain resistant to medications.

We have added reference "9" and we have also added "or myelopathy" line 59. Nikolaidis et al (2010) which was quoted after in the paragraph and thank reviewer 1 for this clarification request. Nikolaidis et al (2010) explain a “low quality evidence that surgery may provide pain relief faster than physiotherapy or hard collar immobilization in patients with cervical radiculopathy; but there is little or no difference in the long-term”. We confirm that we do not mention any lack of interest in surgery for patients with cervical radiculopathy in cervical disk herniations with pain resistant to medications. We thank reviewer 1 for his clarification.

- Material and methods. Was cervical radiculopathy diagnosed just with clinical examination, without performing MR or CT imaging to patients? No specific radiological control before submitting them to cervical traction? Really? It appears, at least, risky.

We thank reviewer 1 for this comment. Patients were diagnosed according to one of the proposals of Romeo et al (2018). In their meta-analysis, they select studies where patients meet these criteria: 1) diagnosis based on magnetic resonance imaging, electromyography, or nerve conduction velocity testing, 2) positive results on at least 3 out of 4 tests, according to clinical prediction rules,21 or 3) symptoms associated with CR, such as pain radiating to the arm, with or without motor or sensitive dysfunction. We have used criteria 3) in this study. To confirm absence of risk, like explain line 159, we performed systematically a cervical spine x-rays

- No neurosurgeons nor ortophaedic surgeons appeared to be involved in the study. Hence, how can patients be informed about their condition and supposed diagnosis? From the reading of this work it seems that, simply, patients with cervical radiculopathy that come to your clinic are directly submitted to cervical traction, without any diagnosis nor alternative treatment proposal; If it’s not, re-write the section entirely, please;

Indeed, we understand the issue. In our health care system, the intervention of neurosurgeons is second line. Diagnostic confirmation is provided by rheumatologists and cervical traction is often performed in rheumatology or physical medicine rehabilitation departments.

- English written language may beneficiate from a mother-tongue speaker review.

We are surprised by this remark. The document has been proofread by Johanna Robertson PT PhD, professional medical and science translator and native British speaker, as evidenced by the attached document. We have written an acknowledgement to this effect (line 437). A new proofreading was done in order to improve manuscript as requested.

 

Reviewer #2:

To Whom It May Concern,

This is an exceptional paper. However, there are a few data points that would add to the paper. Those points are a graphic (histogram) representation of the age breakdown in intervals. We know that older patients are more prone to degenerative spondylosis however, it would be helpful to know where the majority of the study population resides in age distribution. Also, in the methods section I could not discern the exact method used to apply traction. For example, did you all use Gardner Wells Tongs or strap system? Otherwise, well written paper.

We thank reviewer 2 for his compliments and suggestions for improvement.

We have added a graphical table in order to have a better representation of the age of the patients.

We have also added a picture (fig 1) of the device in Methods section to explain the proposed tractions.

 

Reviewer #3:

First, I would like to express sincere gratitude to get an opportunity to review the manuscript. The endeavor of the authors is appreciated. The authors have studied effect of intensive cervical traction with respect to mid-term disability and pain in patients with cervical radiculopathy. However, there is some scope for its improvement.

Specific comments:

1. There are some contradictory statements in the paper. For example, as per methods section 36 patients were enrolled in the study whereas as per results section of main text 42 patients were enrolled. Kindly explain.

We thank reviewer 3 for this comment. This is an error on our part, there are 36 patients included in this work. We have made the correction on line 26 and line 250 and please apologize to the reviewers for this error. 

2. The study design is not clear. Was there any control arm? The conclusion states there was some comparison done. The details of study design need to be well described.

We thank reviewer 3 for this question. There is no control arm. The study is observational, as explained in the title and method. We have added the sentence "All patients received the same treatment" to line 136.

3. Kindly define mid-term effect and intensive cervical traction.

To explicit “mid-term”, we have changed the phrase "at three month" to "at mid-term" line 31. We have also made this clarification on line 197.

 To explicit an intensive cervical traction, we added : "To propose an intensive cervical traction protocol" just before sentence "thirty-minute traction sessions were performed twice daily for 5 days (total of 10 sessions) by a physiotherapist" ligne 162. We thank Reviewer 3 for requesting these clarifications.

4. It would be better if study design were clearly mentioned in title, abstract and main text.

We thank reviewer 2 for this remark. Nevertheless, we have mentioned in the title, abstract and main text the "observational" character of our work. We hesitated with the term "cohort study" which seemed less appropriate for a number of subjects below several hundred.

5. Methods sections is supposed to be core of any study. Here, methods section contains inadequate information. For example, following components for methods section need to be well described.

i. Details of Study design

ii. Setting

iii. Sampling technique

iv. Participant

v. Primary and secondary outcome variables with working definition

vi. Intervention/issue of interest (exposure)

vii. Comparison (if done)

viii. Ethics and end point

ix. Statistical analysis

We thank reviewer 2 for this remark. A formatting error has been corrected to clarify this point. We have taken up several missing headings suggested by Reviewer 3 in order to clarify this legitimate request. In particular, we have added "Details of Study design" line 133, "Intervention/issue of interest (exposure)" line 161, and we have moved the paragraph on ethical approaches to conducting this research to line 226 "Ethics".

6. Kindly re-frame the references as per the guidelines to authors in the home page of journal. Further, most of the studies included in the reference section seem to be the studies published more than 10 years back. If possible, please add recent studies.

We thank proofreader 3 for these remarks. We have changed the bibliography from the "vancouver" style to the "plos one" style available in Zotero followed by a manual check. We understand the remark about studies that may be more than ten years old. Indeed, it is interesting to note how few recent studies on this subject we could find, which motivated our interest.

7. The tables need to be re-organized. For example, table 2 appears first. Further, kindly provide study size in captions. It would be better to frame all the tables in homogenous format.

Table 1 is quoted in the introduction. It therefore seems appropriate to leave it as "Table 1". Nevertheless, we agree with reviewer 3. We have rearranged and modified the tables to make them consistent. 

Section wise comments

1. Abstract contains inadequate information.

a. Mention the design of the study clearly.

b. Methods section is not well described.

We have added the sentence " All patients underwent the same treatment " to clarify the abstract line 13.

c. Kindly start results section with baseline information of participants.

We have made this change and thank you for it.

2. The introduction is lengthy. Please DELETE INFORMATION UNRELATED TO OBJECTIVE so that the section is short and sweet. Kindly shorten this section and delete unrelated information. Kindly focus on three elements of introduction

a. What is known about the topic? (Background)

b. What is not known? (The research problem)

c. Why the study was done? (Justification)

The introduction has been revised to meet the expectations of reviewers 3 and 4. We thank them for their advice.

3. Methods section determines the results. Kindly follow the checklist. Further, kindly focus on three basic elements of methods section.

a. How the study was designed?

b. How the study was carried out?

c. How the data were analyzed?

We hope that the new layout and additions will meet the reviewer's expectations3

4. The discussion section needs to be described scientifically. Kindly frame it along the following lines:

a. Main findings of the present study

b. Comparison with other studies

c. Implication and explanation of findings

d. Strengths and limitations

e. Conclusion, recommendation and future directions

We thank reviewer 3 for this feedback. We have added the suggested headings and redesigned this section to make it easier for practitioners to read.

5. Conclusion needs to provide answers for each objective clearly in a sentence or two.

We thank reviewer 3 for this suggestion and have added " In addition, cervical and radiating arm pain, pain with imagined movements and neuropathic pain also improved " line 429.

 

Reviewer #4:

 Journal of PLOS ONE

27 April 2021

Manuscript No: PONE-D-21-09986_reviewer

“Effect of an intensive cervical traction protocol on mid-term disability and pain in patients with cervical radiculopathy: an exploratory, prospective, observational pilot study”

Dear editorial Teams

In my opinion the manuscript has well written. However, I have several suggestions that I think would improve the manuscript:

a) The Introduction and the Discussion sections are long and redundant.

We have taken this comment into account and thank reviewer 4. We have amended the text accordingly for the introduction. We have redesigned the discussion to make it slightly shorter, but also more reader friendly.

b) Additional descriptions as NDI (line 155- 162) should be summarized, and ets.

We thank Reviewer 3 for this remark. We have taken his advice and deleted several parts in the outcomes description that were not essential to the understanding of the study.

c) Researchers stated that “ forty-two patients were included” and “Only 29 patients could be evaluated at M3: 3 were unavailable as they were undergoing surgery and 4 were lost to follow-up” 42= 29+ 3+ 4+ X? please explain the other 6 patients.

We thank reviewer 4 for this remark which is in line with that of reviewer 3. This is an error on our part, there are 36 patients included in this work. We have made the correction on line 35 and 250. We apologize for this error.. 

d) Please, explain the type of traction used.

We thank reviewer 4 for this comment which is in line with reviewer 2. We have added a picture (fig1 ) to be more explicit about the type of traction.

We sincerely thank the reviewers for their time in reviewing our work. We feel that this feedback has been productive and has helped to improve the communication proposal for our work. We hope we have met their expectations.

 For the authors,

 Dr Thomas Rulleau PT PhD

---

## [Decision Letter · Decision Letter 1]

22 Jul 2021

PONE-D-21-09986R1

Effect of an intensive cervical traction protocol on mid-term disability and pain in patients with cervical radiculopathy: an exploratory, prospective, observational pilot study

PLOS ONE

Dear Dr. Rulleau,

Thank you for submitting your manuscript to PLOS ONE. After careful consideration, we feel that it has merit but does not fully meet PLOS ONE’s publication criteria as it currently stands. Therefore, we invite you to submit a revised version of the manuscript that addresses the points raised during the review process.

If applicable, we recommend that you deposit your laboratory protocols in protocols.io to enhance the reproducibility of your results. Protocols.io assigns your protocol its own identifier (DOI) so that it can be cited independently in the future. For instructions see: http://journals.plos.org/plosone/s/submission-guidelines#loc-laboratory-protocols. Additionally, PLOS ONE offers an option for publishing peer-reviewed Lab Protocol articles, which describe protocols hosted on protocols.io. Read more information on sharing protocols at https://plos.org/protocols?utm_medium=editorial-emailutm_source=authorlettersutm_campaign=protocols.

We look forward to receiving your revised manuscript.

Kind regards,

Panagiotis Kerezoudis, M.D., M.S.

Academic Editor

PLOS ONE

Journal Requirements:

Reviewers' comments:

Reviewer's Responses to Questions

**Comments to the Author**

1. If the authors have adequately addressed your comments raised in a previous round of review and you feel that this manuscript is now acceptable for publication, you may indicate that here to bypass the “Comments to the Author” section, enter your conflict of interest statement in the “Confidential to Editor” section, and submit your "Accept" recommendation.

Reviewer #2: All comments have been addressed

Reviewer #3: (No Response)

Reviewer #4: All comments have been addressed

2. Is the manuscript technically sound, and do the data support the conclusions?

Reviewer #2: Yes

Reviewer #3: Partly

Reviewer #4: Yes

3. Has the statistical analysis been performed appropriately and rigorously? 

Reviewer #2: Yes

Reviewer #3: Yes

Reviewer #4: Yes

4. Have the authors made all data underlying the findings in their manuscript fully available?

Reviewer #2: Yes

Reviewer #3: Yes

Reviewer #4: Yes

5. Is the manuscript presented in an intelligible fashion and written in standard English?

Reviewer #2: Yes

Reviewer #3: Yes

Reviewer #4: Yes

6. Review Comments to the Author

Reviewer #2: (No Response)

Reviewer #3: The paper is in better shape. Congratulations to the authors for such a nice work. Yet, some modifications need to be made before publication. Kindly refer to the following aspects.

1. Conclusion needs to provide answers for each objective clearly in a sentence or two.

a. Here the conclusion does not support the findings of the paper.

b. As the study was not comparative study, such statements need to be removed.

c. The conclusion should only contain information pertaining to findings of the current paper (not other studies).

2. Abstract contains inadequate information.

a. Mention the design of the study clearly.

b. Methods section is not well described.

3. Please DELETE INFORMATION UNRELATED TO OBJECTIVE so that the section is short and sweet. Kindly shorten this section and delete unrelated information. Kindly focus on three elements of introduction

a. What is known about the topic? (Background)

b. What is not known? (The research problem)

c. Why the study was done? (Justification)

4. There are too many strikethrough marks. Kindly delete the text completely which has been asked to delete. At the same time, please highlight the changes that have been made.

5. It would be better to shorten the paper. It seems there is too much of information.

6. Kindly mention tables and figures in line with recommendations of journal. For example, table is not required in introduction.

Reviewer #4: Thank you for your efforts. All comments have been addressed. In this study the effect of cervical traction on neuropathic pain in the case of cervical radiculopathy was assessed.

7. PLOS authors have the option to publish the peer review history of their article (what does this mean?). If published, this will include your full peer review and any attached files.

Reviewer #2: No

Reviewer #3: **Yes: **Dr. Satish Prasad Barnawal

Reviewer #4: No

---

## [Author Response · Author response to Decision Letter 1]

26 Jul 2021

We thank the reviewers for their positive comments on our work. We hope that this version will fully satisfy them.

Reviewer #2: All comments have been addressed

Reviewer #3: (No Response)

Reviewer #4: All comments have been addressed

We thank the reviewers 2 for their validation of our comments and our new version

Reviewer #2: (No Response)

We thank the reviewer 2 for his validation of this new version

Reviewer #3: The paper is in better shape. Congratulations to the authors for such a nice work. Yet, some modifications need to be made before publication. Kindly refer to the following aspects.

we thank reviewer 3 for this compliment

1. Conclusion needs to provide answers for each objective clearly in a sentence or two.

a. Here the conclusion does not support the findings of the paper.

b. As the study was not comparative study, such statements need to be removed.

c. The conclusion should only contain information pertaining to findings of the current paper (not other studies).

We thank reviewer 3 for these remarks. We have modified the document according to this advice. We have removed the reference to the comparison of other studies line 362-364.

2. Abstract contains inadequate information.

a. Mention the design of the study clearly.

b. Methods section is not well described.

We thank reviewer 3 for this analysis. We have added the words "prospective and open" just before “observational study” line 26. To improve clarity, we have also modified the following paragraph by adding the main objective of the study. Finally, we have rewritten the following sentence to make it clearer (line 29-32).

3. Please DELETE INFORMATION UNRELATED TO OBJECTIVE so that the section is short and sweet. Kindly shorten this section and delete unrelated information. Kindly focus on three elements of introduction

a. What is known about the topic? (Background)

b. What is not known? (The research problem)

c. Why the study was done? (Justification)

We understand this remark less. The text has been deeply reworked during our last submission in order to answer this request. It is, in fact, extremely short by keeping only the essential useful to physicians and physiotherapists involved in these tractions. We understand that other expert actors may want an even shorter text, but it seems useful to us here to keep the remaining elements to contextualize.

4. There are too many strikethrough marks. Kindly delete the text completely which has been asked to delete. At the same time, please highlight the changes that have been made.

We sent two versions, as requested by the publisher, one with the marks, the other without, to clarify the reviewers' re-reading of the changes. We have, of course, respected this request at every stage.

5. It would be better to shorten the paper. It seems there is too much of information.

We thank reviewer 3 for this comment. We have streamlined our text to meet this request.

6. Kindly mention tables and figures in line with recommendations of journal. For example, table is not required in introduction.

We understand the comment of reviewer 3. We had quoted the table in the introduction. In accordance with the recommendations to the authors, we have therefore illustrated with the table. We have made the modification requested by reviewer 3 and the table now appears in the discussion when it is quoted a second time in the text.

Reviewer #4: Thank you for your efforts. All comments have been addressed. In this study the effect of cervical traction on neuropathic pain in the case of cervical radiculopathy was assessed.

We thank the reviewer 4 for this compliment and validation of this new version

---

## [Editor Report · Decision Letter 2]

28 Jul 2021

Effect of an intensive cervical traction protocol on mid-term disability and pain in patients with cervical radiculopathy: an exploratory, prospective, observational pilot study

PONE-D-21-09986R2

Dear Dr. Rulleau,

We’re pleased to inform you that your manuscript has been judged scientifically suitable for publication and will be formally accepted for publication once it meets all outstanding technical requirements.

Kind regards,

Panagiotis Kerezoudis, M.D., M.S.

Academic Editor

PLOS ONE
---

## [Editor Report · Acceptance letter]

30 Jul 2021

PONE-D-21-09986R2 

Effect of an intensive cervical traction protocol on mid-term disability and pain in patients with cervical radiculopathy: an exploratory, prospective, observational pilot study 

Dear Dr. Rulleau:

I'm pleased to inform you that your manuscript has been deemed suitable for publication in PLOS ONE. Congratulations! Your manuscript is now with our production department. 

Kind regards, 

on behalf of

Dr. Panagiotis Kerezoudis 

Academic Editor

PLOS ONE